# Tracking daily activities with ecological momentary assessment: A bibliometric analysis of current use in health Mapping daily activities with ecological momentary assessment

Eva Delooz[1]*, Bruno Bonnechère[1,2], Barbara Piškur[1,3], Annemie Spooren[1,2]

**1** Research Center REVAL, Hasselt University, Diepenbeek, Belgium, **2** Centre of Expertise in Care Innovation, Department of PXL-Healthcare, PXL University of Applied Sciences and Arts, Hasselt, Belgium, **3** Zuyd University of Applied Sciences, Heerlen, The Netherlands

* eva.delooz@uhasselt.be

## Abstract

Health research has shifted from a disease-centered approach towards emphasizing functioning and more specific lived health. Lived health, the actual performance of daily activities in one's environment, has nevertheless received limited attention, and its assessment remains methodologically challenging. Ecological Momentary Assessment (EMA), a real-time method capturing behaviors, emotions, and context in natural settings, holds promise in this regard. Although EMA research is increasing, insight into its use for studying daily activities is still limited. To address this gap, this study systematically maps EMA applications across diverse health and disability populations to better understand lived health, defined as actual engagement in daily activities. A bibliometric analysis was conducted using a literature search on Web of Science with keywords related to EMA combined with daily activity terms, yielding 3,692 English-language articles. Publications were classified according to general characteristics, distribution of disability and health populations, actual engagement in daily activities following the person-environment-occupational model (PEO-model), and interaction analyses combining the last two analyses. The results show that mental disorders dominate the EMA research on daily activities, representing 75% of the dataset, which has significantly shaped the overall research landscape. Moreover, while personal factors are frequently highlighted, occupational and environmental dimensions remain underrepresented. These findings suggest that future EMA research should better integrate aspects of person, occupation, and environment, for instance by using tools such as geolocation and passive sensing to capture daily functioning more holistically. Expanding research beyond mental health and increasing secondary analyses will further strengthen the relevance and impact of EMA on health research.

**Data availability statement:** All materials necessary to reproduce the analyses are publicly available via Zenodo (DOI: 10.5281/zenodo.18163870). The repository includes (i) all R scripts used for data processing and analysis, (ii) detailed R version and package information (R version 2024.12.0+467), (iii) the complete search strategy and search terms, and (iv) a comprehensive README file with step-by-step instructions to reproduce the analyses. Due to licensing restrictions imposed by Web of Science (Clarivate Analytics), the raw bibliographic records underlying this study (n = 3,692 articles) cannot be shared publicly. However, all information required to reproduce the data collection process from Web of Science is provided in the repository. Researchers with institutional access to Web of Science can therefore fully reproduce the dataset by following the documented search strategy.

**Funding:** This research was supported by a doctoral research grant from Hasselt University awarded to the principal investigator (ED), grant number BOF24DOC43. No other financial or material support was received for this study. The funder had no role in study design, data collection and analysis, decision to publish, or preparation of the manuscript. No authors received a salary or other financial compensation specifically for the conduct of this study.

**Competing interests:** The authors have declared that no competing interests exist.

## Author summary

In our research, we explored how people's everyday activities are studied in disability and health populations. Traditionally, the focus was on the disease, but there is growing recognition that it is just as important to understand how people engage in their daily lives, (i.e., functioning). We looked at a method called "ecological momentary assessment," which involves collecting information from people in real time as they go about their day. This approach can provide unique insights into how health affects daily functioning in natural settings. To see how this method is currently used, we examined 3,692 English-language articles. We found that most of the research has been done in the field of mental health, while other areas - such as the role of the occupation and the environment - receive far less attention. This imbalance means that we still have only a partial picture of how disability and health conditions affect everyday activities. Our study shows the need for broader use of real-time approaches to capture daily life across different disability and health conditions and contexts. By expanding the focus beyond mental health and including factors such as where people are and what they are doing, future research can provide a more complete and realistic understanding of daily functioning.

## 1 Introduction

Since 1948, the understanding of health has evolved from a static ideal of complete well-being to a dynamic balance of physical, mental, social, and existential well-being, requiring continuous adaptation to life's changing conditions and environments [1]. As a result, health research has undergone a paradigm shift—from disease-centered models to frameworks that prioritize functioning [2]. Grounded in the World Health Organization's International Classification of Functioning, Disability and Health (ICF), the concept of functioning extends beyond biological health (i.e., bodily structures and functions) to include lived health: the actual performance of daily activities within one's environment [3]. Functioning is increasingly recognized as the crucial bridge linking health not only to individual well-being but also to broader societal welfare. This paradigm shift fosters a more comprehensive understanding of health's value, supporting global goals like the United Nations' Sustainable Development Goal 3 (SDG 3): Good Health and Well-being [2,4,5].

This evolving focus on functioning and lived health naturally directs attention to the role of daily activities, which are no longer viewed as routines, but as active determinants of health [6]. The performance of daily activities reflects the dynamic interplay between personal capacity, environmental support, and participation, core elements of the ICF framework [3]. Although the ICF identifies critical components of this relationship, it provides limited practical guidance about how these elements dynamically interact to facilitate daily activities [7]. The Person-Environment-Occupation (PEO)

model addresses this gap by offering a visual Venn-diagram representation of the dynamic relationship of person, environment, and occupation to understand daily activities [7–11].

Although daily activities are conceptually framed as the dynamic interaction between person, environment, and occupation, no existing assessment fully captures this complexity in a real-life context. Some tools reflect aspects of this interplay: for example, the Canadian Occupational Performance Measure (COPM) offers valuable insight into individuals' everyday experiences [12]. However, most current methods rely on retrospective self-reports and observations, which are limited in their ability to detect the subtle, moment-to-moment fluctuations that shape daily activities [13,14]. Because these methods look back on experience rather than capturing it as it unfolds, they are prone to memory bias, social desirability, and inaccurate reconstruction. As a result, they fail to access the microprocesses, small but meaningful behavioral and cognitive patterns, that influence functioning and lived health [11,15]. Two key challenges emerge: (1) the absence of an assessment that directly measures the dynamic interplay between PEO components, and (2) the inability of current tools and assessments to capture the microprocesses of the dynamic interplay between the PEO components.

To address these challenges to capture functioning and lived health through daily activities Ecological Momentary Assessments (EMA) could be promising by capturing real-time behavioral, affective, and contextual data in natural settings [16]. EMA also known as ambulatory assessment or experience sampling methodology, is widely used in health sciences, psychology, and clinical research to assess symptoms, monitor treatment, and to develop personalized interventions such as Just-In-Time Adaptive Interventions (JITAIs) for individuals with or without disabilities [17–21]. While EMA has gained substantial traction in health research, existing reviews have predominantly focused on functioning as a biological aspect of health, with the focus on diagnoses or specific symptoms [17–21]. Another major focus has been on the methodological characteristics of EMA, such as sampling strategies, compliance rates, and data analysis techniques [22,23]. Recent reviews have examined the use of EMA in health research, with a primary focus on mental health applications, mobile-based interventions, and methodological characteristics such as sampling strategies and compliance [24,25]. While these studies provide valuable overviews of EMA adoption and technological development, they predominantly conceptualize functioning in terms of symptoms, diagnoses, or intervention outcomes.

Although EMA holds promise for assessing functioning and lived health by capturing the dynamic microprocesses of daily activities in real time, its use for this purpose remains limited. Current EMA research has primarily focused on symptom monitoring or methodological refinement, while the broader potential of EMA to explore how individuals engage in and make meaning of everyday activities has been underexplored. To address this gap, the present study aims to systematically map the use of EMA to understand lived health, defined as actual engagement in daily activities, guided by the PEO-model.

## 2 Materials and methods

A bibliometric analysis was conducted to systematically map the published literature on EMA use in studying lived health and daily activities in disability and healthy populations. This method enables quantitative evaluation of publication trends, thematic clusters, methods, and gaps [26]. The BIBLIO checklist is available in the supplementary documents.

### 2.1 Data-collection

A data search was conducted in Web of Science on February 17, 2025, using the following search terms: ("ecological momentary assessment" OR "ecological momentary" OR "ambulatory assessment" OR "experience sampling" OR "real time assessment" OR "Behavioral data collection") AND ("daily functioning" OR "daily activities" OR "daily activity" OR "activities of daily living" OR "occupational performance" OR "occupation" OR "activity" OR "task" OR "physical activity" OR "leisure" OR "self-care" OR "play" OR "productivity" OR "work" OR "participation"). These search terms were refined during an initial scoping phase to balance sensitivity and precision to improve the quality of the search. Web of Science was selected as the source database due to its standardized bibliographic structure, inclusion of Keywords Plus and

subject categories, and suitability for reproducible bibliometric analyses across large publication corpora. Only peer-reviewed English publications were analyzed. The search yielded 3,692 included publications. The detailed information of all selected publications was downloaded in BibTeX format.

## 2.2 Data-analysis

The data-analysis was structured into four main components using R software version 2024.12.0+467: (a) an analysis of the general characteristics (b) an analysis of the distribution of disability and health populations, (c) an analysis of lived health or engagement in daily activities according to the PEO model and (d) an interaction analysis combining the second and third component. For the first component, the analysis focuses on publication output and growth over time, including the number of publications per year. Additionally, it examines the types of publications (e.g., empirical, review, method-ological) and the countries of origin, to provide an overview of how and where EMA is being used in the context of functioning and lived health.

The second component was guided by the Global Burden of Disease (GBD) classification, using tailored search terms for non-communicable diseases and disability-related conditions. The GBD framework was selected for its standardized disability metrics of Disability Adjusted Life Years (DALYs), policy relevance, and methodological transparency [27]. To include non-disease populations, an additional category 'healthy living' was added, covering studies on healthy individuals, students, and adults without diagnoses (term list available on Zenodo with DOI 10.5281/zenodo.18163870).

For the third component following the health-based classification of lived health, a parallel categorization was conducted using the PEO model. The included publications were classified according to the PEO's three core domains: person (affective, cognitive, physical components), occupation (self-care, productivity, leisure) and environment (physical, institutional, cultural, social contexts) [8–10]. The complete term list can be found on Zenodo with DOI 10.5281/zenodo.18163870.

Finally, the fourth component integrates the second and third analysis. Percentages distribution was calculated to assess the distribution of PEO components across the various disability and health categories. This methodological choice ensured that larger research domains exerted a proportional influence on aggregate trends, while smaller fields contributed in line with their respective sample sizes. By applying this approach, the analysis focusing on mitigated the risk of overemphasizing findings from niche or less-represented areas and provided a balanced overview of dominant research paradigms within EMA studies related to daily activities and functioning.

The classification of the publications throughout the different components, relied on a keyword-matching procedure implemented in R. Predefined lists of stemmed keywords were matched against multiple bibliographic fields, including titles (TI), abstracts (AB), author keywords (DE), Keywords Plus (ID), Web of Science categories, and subject categories. These fields were concatenated into a single searchable text string for each publication. Each publication was assigned to a single category based on the first detected keyword match. Multiple category assignments were therefore not permitted. If no match was identified in the combined fields, additional matching was performed sequentially on author keywords and titles only. The classification process was fully algorithmic, and individual publications were not manually reassigned. To improve coverage, the keyword lists were iteratively refined by inspecting records that remained unclassified and identifying recurring relevant terms, after which the classification procedure was rerun until no further systematic assignments could be made.

## 2.3 Data-availability

All R scripts, search terms, and documentation necessary to reproduce the analyses, including R version information (R version 2024.12.0+467), are available in a public Zenodo repository with DOI 10.5281/zenodo.18163870. The underlying bibliographic data were obtained from Web of Science (Clarivate Analytics) and cannot be shared publicly due to licensing restrictions.

# 3 Results

## 3.1 Publication output and growth rate

This bibliometric analysis encompasses 3,692 documents published between 1987 and 2025. The field has demonstrated remarkable growth, particularly from 2018 onward, with annual publications exceeding 400 by 2022 and approaching 600 in 2024 (Fig 1). Because data-extraction was done on February 17, 2025, the year 2025 (n = 70 available publications at the time of data-collection), could not be considered as a full year and therefore was excluded in Fig 1.

The top five of dominating countries publishing on EMA related to engagement in daily activities is: the United States dominating 43.4% of publications, followed by Germany (10.2%), the Netherlands (6.7%), China (6.5%), and the United Kingdom (5.2%). The corpus draws from 1,387 diverse journals, books, etc. and cites 108,303 references. Original research articles constitute the majority (3,145 publications), complemented by 144 reviews and 403 other document types (including clinical trials and case reports).

## 3.2 Disability and health categories

The distribution of research within the domain of EMA focusing on daily activities across different health and disability populations reveals significant trends in the focus of current studies. 201 Publications were identified as falling outside the health and disability focus areas, predominantly covering topics such as energy sustainability and computer science environmental studies. These publications were removed from the data file and excluded from further analysis (Fig 2).

As illustrated in Fig 3, mental disorders emerged as the predominant research focus with 2,501 publications (representing 75% of the total dataset). This was followed by studies on healthy populations (517 publications, representing 14.8% of the total dataset), while neurological and brain disorders (88 publications, representing 2.5% of the total dataset), cardiovascular conditions (86 publications, representing 2.4% of the total dataset), and cancer diseases (72 publications, representing 2.1% of the total dataset) constituted smaller yet significant research clusters.

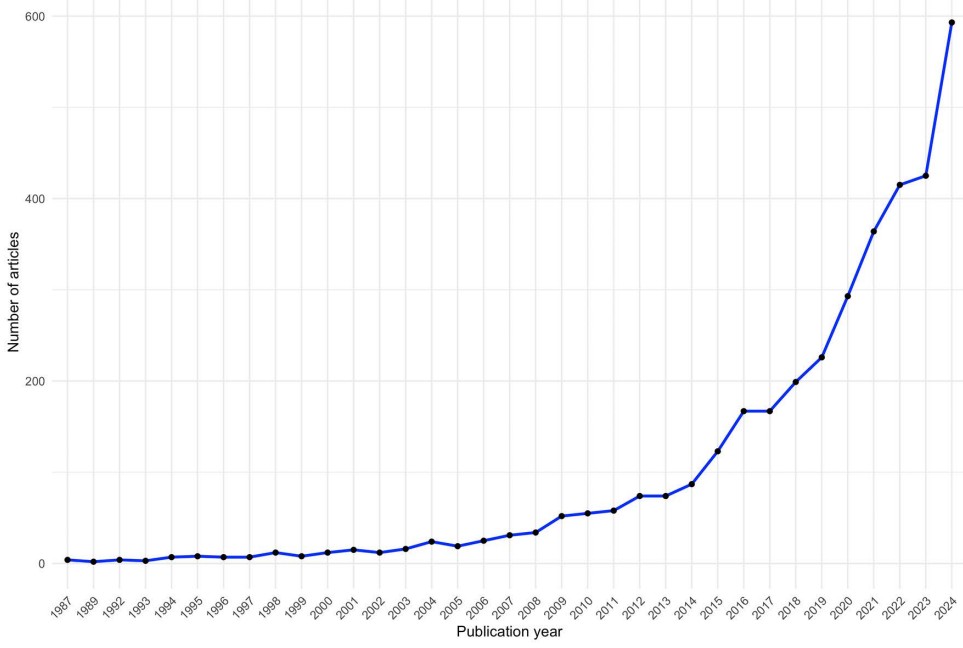

**Fig 1. Number of publications per year.**

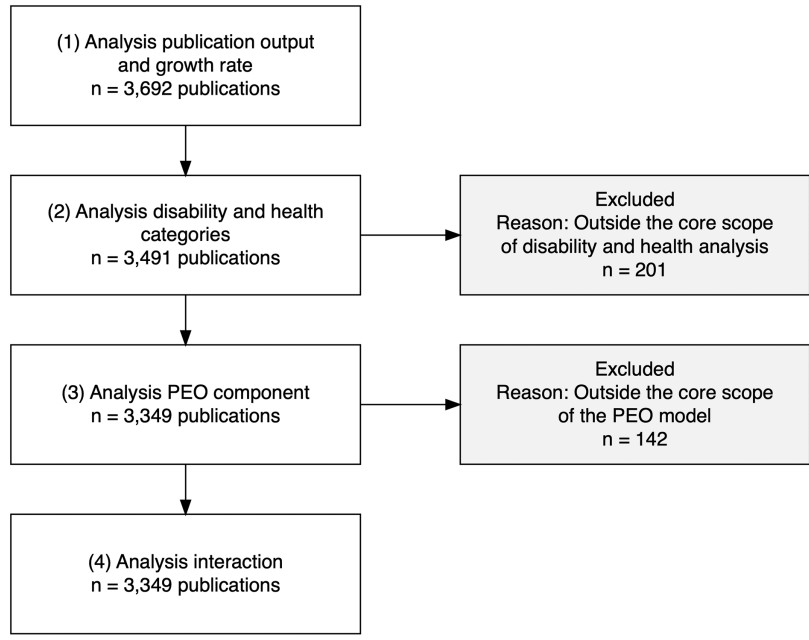

**Fig 2. Flowchart.**

### 3.3 PEO components

The PEO component analysis was based on the 3,491 publications that remained after categorizing studies by health and disability (Fig 2). An additional 142 publications were excluded because they focused on biomedical or applied physical research, validation studies, or protocols that fall outside the core scope of the PEO model.

The results focusing on 3,349 publications reveal a clear imbalance in how EMA is applied within the PEO framework. As shown in Fig 4, person-oriented research dominates the field, with 2,254 publications (61.1%), compared to substantially fewer studies addressing occupation (659 publications, 17.8%) and environment (321 publications, 8.7%).

Although the PEO model is typically represented as a balanced, interconnected framework (Fig 5a), current EMA applications heavily favor personal factors. In fact, 61.1% of the publications focuses primarily on the 'person' component, deviating from the model's intended integration of person, occupation, and environment (Fig 5b). While numerical distributions are reported in the text, Fig 5a and 5b serves solely as a conceptual visualization and does not depict relative magnitudes.

### 3.4 Interaction between disability and health categories and PEO components

Analysis of 3,349 publications (Fig 2) shows distinct patterns in research focus across disability and health categories when mapped to the CMOP-E components (Fig 6). Three primary trends emerge from the data, along with some notable exceptions that warrant attention.

The first and most prominent pattern shows a strong person-centered focus (60% or more) in several key domains, such as digestive diseases at 75%, liver diseases at 66.7% and mental disorders with 63.2% person-focused studies. A second clear pattern emerges in occupation-dominant research (55% or more), particularly evident in musculoskeletal disorder investigations (62.5%), diabetes and kidney disease research (58.9%) and healthy living studies (56.1% occupation focus). The third major pattern appears in environment-driven research (50% or more), most strikingly in studies of infectious diseases (63.6%) and respiratory diseases (53.1% environment focus).

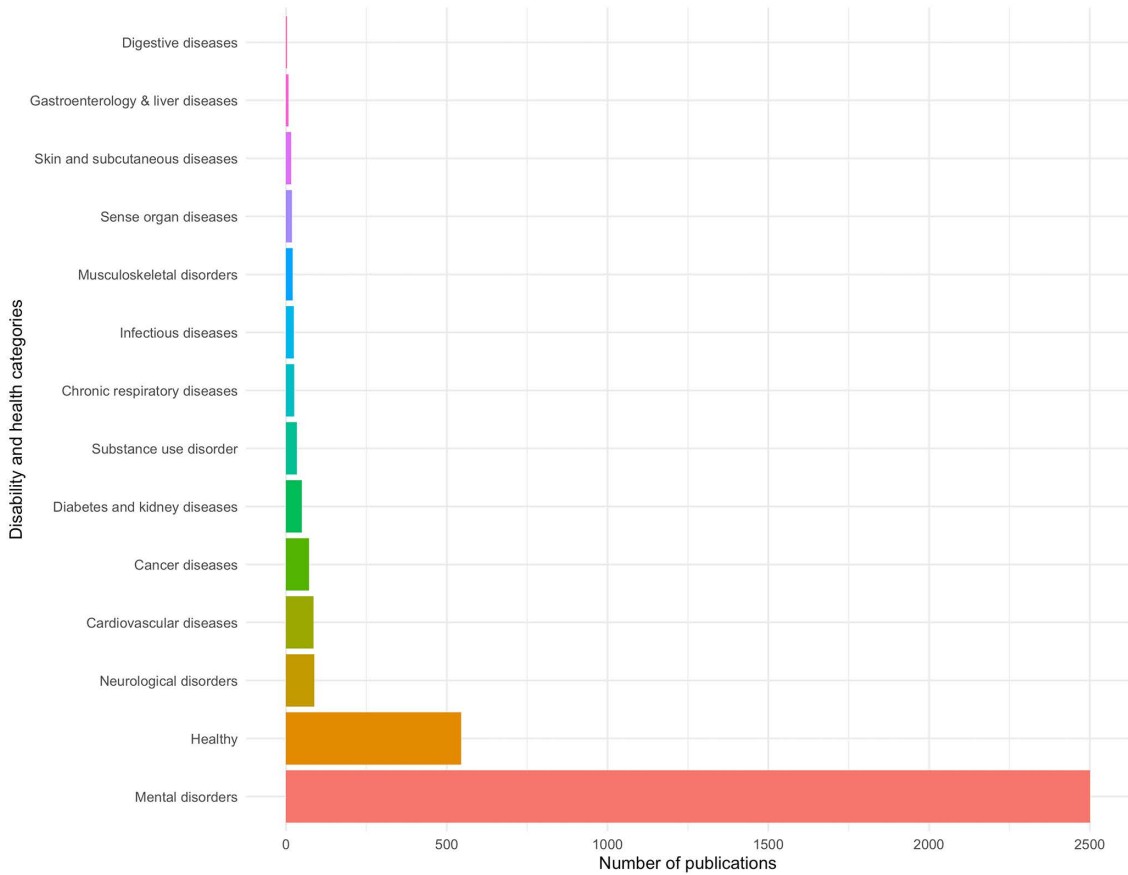

**Fig 3. Number of publications per disability and health categories.**

Several health domains present more balanced distributions that deviate from these primary patterns. Sense organ diseases show remarkable equilibrium between environment and occupation components (45% each), while skin diseases research divides evenly between person and occupation factors (50% each). At the extremes of distribution, some notable gaps appear. Digestive disease research shows no attention to occupation components, while skin disease studies similarly marginalize environmental factors.

## 4  Discussion

This study aimed to map how EMA is used to explore lived health experiences and engagement in daily activities across disability and healthy populations, guided by the PEO model. A bibliometric analysis revealed that EMA research on daily activities has evolved significantly since its early conceptualization by Stone and Shiffman (1994). Originally limited by 1990s technology (e.g., pagers, electronic diaries), earlier frameworks like Ambulatory Assessment and ESM were already in use before EMA's formal naming in 1994 [16,23]. The advent of smartphones in the late 2000s marked a turning point, enabling "smart EMA" via mobile devices like smartphones, tablets, and wearables [28]. This technological shift accelerated EMA research, driven by widespread digital integration in daily life [24,29].

Using the GBD classification enabled structured comparison across health categories, though such frameworks have limitations, e.g., categorization issues like autism being classified as mental or neurological [30]. While alternative classification systems (e.g., ICF) exist for functional assessment, the GBD was selected for its hierarchical structure and

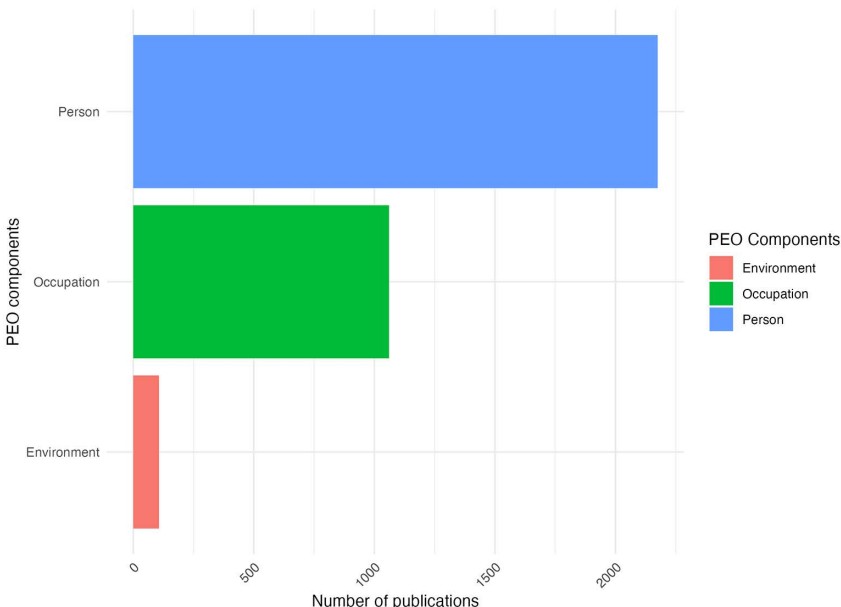

**Fig 4. Number of publications per PEO components.**

**5a Representation of the PEO model**     **5b Results of the PEO components**

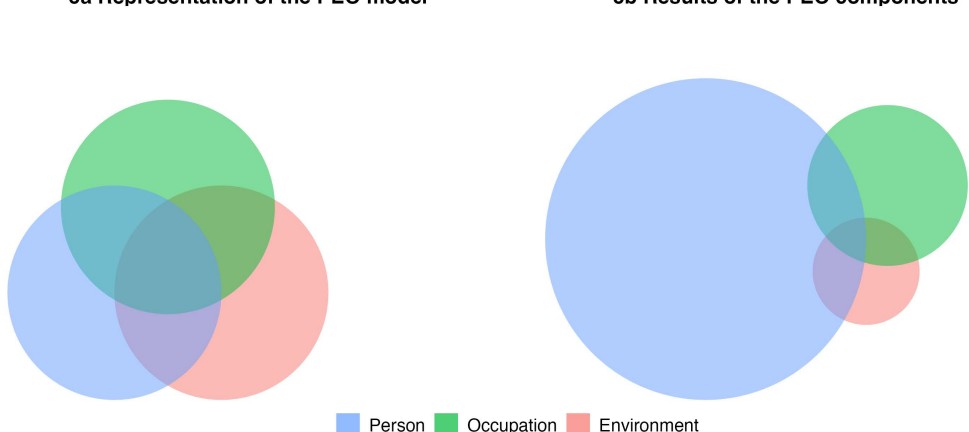

**Fig 5. Representation of the PEO model in relation to the results.**

disease-specific weights, allowing systematic comparisons across categories. However, the selection of any classification framework involves inherent limitations, particularly for conditions that span across multiple domains. Findings revealed a notable mismatch with GBD data, where neurological disorders show much higher population-level impacts than mental disorders: prevalence 35,334 vs. 13,554, and incidence 10,264 vs. 5,460 per 100,000 people [31,32]. However, the results of this bibliometric analysis suggest that mental disorders research dominates the EMA literature regarding daily activities (75% of publications) compared to neurological studies (3%). This imbalance reflects the historical roots of EMA within psychology, where it was originally developed to monitor symptoms, affective states, and behavioral triggers in real time [16]. Importantly, this pattern is consistent with previous bibliometric analyses, which have documented a rapid

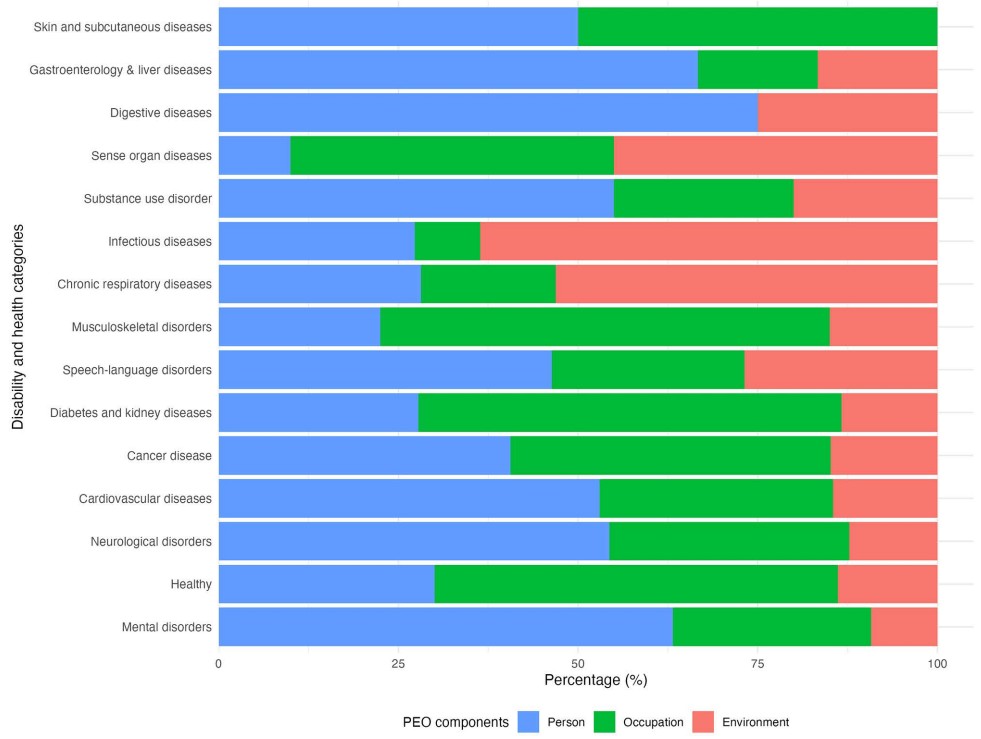

**Fig 6. Interaction between disability and health categories and PEO components.**

growth of EMA publications over the past decade, largely driven by mental health research and accompanied by increasing integration of mobile and sensor-based technologies [24]. While this focus aligns with dominant EMA research trajectories, it simultaneously highlights a substantial underutilization of EMA in neurological contexts - particularly for capturing engagement in daily activities and functioning.

This study uniquely focused on functioning and lived health experiences, aligning with the concept of daily activities. The PEO model conceptualizes daily activities as a dynamic, multidimensional interaction between person, environment, and occupation, described in the literature using terms such as occupation, activity, performance, and participation. While these terms emphasize different aspects, this study adopts the PEO model to represent engagement in daily life as shaped by personal and environmental interactions, central to the idea of "lived health" [2]. Developed within occupational therapy, the PEO model was selected for its explicit focus on daily activity engagement, aligning closely with this study's aim [9,10]. While model selection involves trade-offs, PEO was preferred over broader systems for its conceptual richness and direct alignment with everyday functioning. Although frameworks like the ICF offer broader universality, they insufficiently capture the unique interplay between person, activity, and environment. Stamm noted that while the ICF addresses function and context, it overlooks the subjective experience of occupation, the core of occupational therapy models like PEO [33]. Hammell similarly critiques the ICF for fragmenting occupation-related concepts across domains [7]. Beyond methodological fit, the PEO model provides a rich theoretical base, well aligned with EMA's potential to capture real-time microprocesses of daily functioning [16]. EMA operationalizes what PEO conceptualizes: lived activity in context. Together, they offer a complementary lens for exploring how people engage in daily life, both in general populations and those with complex functional challenges.

The analysis of PEO components shows a disproportionate emphasis on personal factors (affective, cognitive, and physical) which reflects broader trends in healthcare [3]. Person-related factors dominate in mental (63.2%), digestive

(75%), and liver diseases (66.7%), reflecting a psychological and biomedical focus. In contrast, occupation-related research is most prevalent in healthy populations (56.1%), diabetes/kidney (58.9%), and musculoskeletal conditions (62.5%), highlighting the role of self-management and daily routines. Environmental factors feature more in infectious (63.6%) and respiratory diseases (53.1%), aligning with known external influences like pathogens and air quality. However, certain gaps become apparent. For example, the focus on occupation is absent in research on digestive diseases, and environmental factors are poorly addressed in studies on skin conditions. This may suggest methodological or conceptual blind spots that risk narrowing the scope of our understanding of functioning in various health populations.

Beyond its conceptual contributions, these findings have important practical implications. The observed dominance of person-focused EMA studies suggests that current EMA protocols may insufficiently capture the occupational and environmental dimensions of daily functioning. For researchers and clinicians, this highlights the need to design EMA studies that more deliberately integrate activity-related and contextual variables to improve ecological validity and clinical relevance. Incorporating EMA data on daily activities, routines, and environmental context may support more personalized and context-sensitive interventions, particularly in rehabilitation and chronic care settings. Moreover, the identified underrepresentation of EMA applications in several disability and health domains suggests opportunities for broader implementation of EMA to inform practice in populations where daily functioning is substantially affected.

### 4.1 Limitations

While this study provides valuable insights into trends in the use of EMA for assessing daily activities, several limitations must be acknowledged. First, the literature search was conducted exclusively in the Web of Science database, without inclusion of other major databases such as PubMed or Scopus. Consequently, relevant studies indexed elsewhere may have been missed, and the findings are limited to publications available within Web of Science. Second, despite careful construction of the search strategy, the selected keywords may not have fully captured the entire scope of relevant literature. To reduce false positives and maintain a conceptually coherent dataset, the search strategy prioritized precision by restricting queries to the full phrases "ecological momentary assessment" and "experience sampling", rather than the use of acronyms such as EMA or ESM. As a result, some relevant studies using acronyms exclusively may not have been captured; however, this number is expected to be small, as these terms are typically defined in full in abstracts. Consequently, the total volume of EMA research on daily activities and functioning may be slightly underestimated. In addition, although patient-reported outcome measures (PROMs) are conceptually related to EMA, they were not explicitly included as search terms. The present study intentionally focuses on EMA-based approaches, defined by repeated, real-time assessments conducted in naturalistic contexts - key methodological characteristics that are not necessarily or explicitly present in PROM-based research. While PROMs constitute an important body of literature in the assessment of lived health and functioning, they represent a distinct methodological tradition. Consequently, the findings specifically reflect trends within EMA-labeled research rather than patient-reported assessment approaches more broadly. Broader or more specific terminology might have yielded a more comprehensive dataset. Third, the review was restricted to English-language publications, potentially introducing a language bias by excluding studies published in other languages that may offer additional perspectives. Moreover, the analysis represents a snapshot of the field up to February 17, 2025. Subsequent publications and developments are not reflected in this review.

Methodologically, the classification of publications into disability and health domains, as well as PEO components, relied on a keyword-matching procedure implemented in R. While this approach ensured systematic and reproducible categorization across a large corpus of publications, the resulting classifications were inherently shaped by the predefined keyword lists and the availability of information within indexed bibliographic fields (e.g., titles, abstracts, and metadata). Consequently, publications in which relevant concepts were discussed primarily outside these fields may not have been fully captured. To enhance transparency and reproducibility, the complete list of search terms and classification criteria is provided on Zenodo with DOI 10.5281/zenodo.18163870.

## 4.2 Future research

This bibliometric mapping highlights several key directions for future EMA research. First, there is a critical need to integrate all three components of the PEO model, person, occupation, and environment, in the EMA. While current EMA studies predominantly focus on personal factors such as affective, cognitive, and physical states, future protocols should also incorporate occupational and environmental dimensions, as their interaction plays a crucial role in shaping functioning and daily activity engagement. For example, the use of geospatial tracking, passive sensing, and context-aware mobile applications can offer valuable insights into how environmental factors (such as location, air quality, or social context) dynamically interact with individuals' occupational performance and personal states. Capturing this interplay is crucial for understanding the complexity of daily functioning and could lead to the development of more targeted and effective rehabilitation programs and interventions. Second, there is a need to expand research beyond the currently dominant disability and health domains. EMA studies have largely focused on mental health. Broadening the scope to include a wider range of disability and health conditions would create a more inclusive and globally relevant evidence base. Such expansion could uncover unique daily functioning in patient groups that experience severe problems in engaging in daily life activities. Finally, the field would benefit from a greater emphasis on secondary research, such as scoping reviews, bibliometric analysis, systematic reviews and meta-analyses. The current literature remains heavily skewed toward primary empirical studies, limiting the ability to synthesize findings across diverse contexts. Reviews could critically appraise methodological variations, identify best practices for capturing complex interactions between person, occupation, and environment, and guide future technological innovations in EMA.

## 5 Conclusion

This bibliometric analysis mapped EMA research on daily activities within health and disability populations, using the PEO model as a framework. Findings show a strong focus on personal factors, with limited attention to occupation and environmental context. While EMA's capabilities have grown through technological advances, its potential to capture dynamic person–environment–occupation interactions remains underused. Mental disorders dominate the field, comprising 75% of publications. This reflects EMA's psychological roots. Expanding EMA use to other health domains, such as neurological, cardiovascular, and oncological conditions, could enhance monitoring of lived health, engagement in daily activities, in real-world settings.

## Supporting information

**S1 Checklist. The BIBLIO checklist for reporting the bibliometric reviews of the biomedical literature.** Checklist reporting the methodological details of the bibliometric analysis conducted in this study.
(DOCX)

## Author contributions

**Conceptualization:** Eva Delooz.

**Investigation:** Eva Delooz.

**Methodology:** Eva Delooz.

**Supervision:** Bruno Bonnechère, Barbara Piškur, Annemie Spooren.

**Writing – original draft:** Eva Delooz.

**Writing – review & editing:** Eva Delooz.

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
