## [Decision Letter · Decision Letter 0]

2 Dec 2025

PDIG-D-25-00686Tracking  daily  activities  with  ecological  momentary  assessment:  a  bibliometric  analysis  of  current  use  in  healthPLOS Digital Health Dear Dr. Delooz, Thank you for submitting your manuscript to PLOS Digital Health. After careful consideration, we feel that it has merit but does not fully meet PLOS Digital Health's publication criteria as it currently stands. Therefore, we invite you to submit a revised version of the manuscript that addresses the points raised during the review process. Please submit your revised manuscript by Jan 31 2026 11:59PM. If you will need more time than this to complete your revisions, please reply to this message or contact the journal office at digitalhealth@plos.org.  Please include the following items when submitting your revised manuscript:* A rebuttal letter that responds to each point raised by the editor and reviewer(s). You should upload this letter as a separate file labeled 'Response to Reviewers'. This file does not need to include responses to any formatting updates and technical items listed in the 'Journal Requirements' section below.'. This file does not need to include responses to any formatting updates and technical items listed in the 'Journal Requirements' section below.* A marked-up copy of your manuscript that highlights changes made to the original version. You should upload this as a separate file labeled ''. This file does not need to include responses to any formatting updates and technical items listed in the 'Journal Requirements' section below.'. This file does not need to include responses to any formatting updates and technical items listed in the 'Journal Requirements' section below.* A marked-up copy of your manuscript that highlights changes made to the original version. You should upload this as a separate file labeled 'Revised Manuscript with Track Changes'.'.* An unmarked version of your revised paper without tracked changes. You should upload this as a separate file labeled ''.'.* An unmarked version of your revised paper without tracked changes. You should upload this as a separate file labeled 'Manuscript'.'. If you would like to make changes to your financial disclosure, competing interests statement, or data availability statement, please make these updates within the submission form at the time of resubmission. Guidelines for resubmitting your figure files are available below the reviewer comments at the end of this letter. We look forward to receiving your revised manuscript. Kind regards, Torleif Markussen LundeGuest EditorPLOS Digital Health Ziad El-KhatibSection EditorPLOS Digital Health Leo Anthony CeliEditor-in-ChiefPLOS Digital Healthorcid.org/0000-0001-6712-6626 '.'. If you would like to make changes to your financial disclosure, competing interests statement, or data availability statement, please make these updates within the submission form at the time of resubmission. Guidelines for resubmitting your figure files are available below the reviewer comments at the end of this letter. We look forward to receiving your revised manuscript. Kind regards, Torleif Markussen LundeGuest EditorPLOS Digital Health Ziad El-KhatibSection EditorPLOS Digital Health Leo Anthony CeliEditor-in-ChiefPLOS Digital Healthorcid.org/0000-0001-6712-6626 **Journal Requirements:**

1. Please clarify all sources of funding (financial or material support) for your study. List the grants (with grant number) or organizations (with url) that supported your study, including funding received from your institution.

2. State the initials, alongside each funding source, of each author to receive each grant.

3. State what role the funders took in the study. If the funders had no role in your study, please state: “The funders had no role in study design, data collection and analysis, decision to publish, or preparation of the manuscript.”

4. If any authors received a salary from any of your funders, please state which authors and which funders.

2. We have amended your Competing Interest statement to comply with journal style. We kindly ask that you double check the statement and let us know if anything is incorrect.

3. In the online submission form, you indicated that The data underlying this study are available from the Web of Science database (Clarivate Analytics), subject to their access and licensing terms. The dataset analyzed during the current study, including records of 3,692 articles, and the R scripts used for data processing and analysis, are available from the corresponding author upon reasonable request.

3. Uploaded as supplementary information.

4. Please provide separate figure files in .tif or .eps format.

5. We have noticed that you have uploaded Supporting Information files, but you have not included a list of legends. Please add a full list of legends for your Supporting Information files after the references list.

 **Additional Editor Comments:** Thank you for writing this paper which addresses a gap in the literature; how Ecological Momentary Assessment (EMA) is used to study daily activities and functioning across health populations. I find the combination of global burden of disease categories with the person-environment-occupation model a fresh perspective, and the writing is clear. However, several issues need resolution before publication. I recommend major revision. Thank you for writing this paper which addresses a gap in the literature; how Ecological Momentary Assessment (EMA) is used to study daily activities and functioning across health populations. I find the combination of global burden of disease categories with the person-environment-occupation model a fresh perspective, and the writing is clear. However, several issues need resolution before publication. I recommend major revision.

Major concerns requiring action

PLOS Digital Health requires open data deposit with a permanent identifier. Your current statement, “available from the corresponding author upon reasonable request”, does not meet journal policy. Upload your processed dataset or script to replicate data extraction (not raw Web of Science exports, which are proprietary), term lists, and R scripts to a suitable data deposit. Provide the DOI in your Data Availability Statement. Include R package versions for reproducibility.

Your analysis classified publications using keyword matching of titles, abstracts, and metadata. As I understand you did not examine actual EMA protocols or study designs. You describe this approach Lines 324-329 describe this approach, but the classification process needs fuller methodological detail. The methods mention using an R 'list' function with keywords in supplementary materials, but several specifics remain unclear: Which bibliographic fields did you search (title only, or title + abstract + keywords)? Could one publication be assigned to multiple categories? Was classification purely automated, or did you manually verify any assignments? If purely algorithmic, acknowledge that keyword matching cannot capture nuanced content. A paper discussing depression and physical activity in its results section might be misclassified if those terms appear only there, not in the title or abstract you searched.

Your search was limited to Web of Science using full phrases ("ecological momentary assessment," "experience sampling") but excluded the acronyms EMA and ESM. Many EMA papers use only acronyms in titles and abstracts, creating risk of missed relevant studies. I recognize these abbreviations appear in other fields, which may have deterred their inclusion. However, a pilot search combining acronyms with your EMA-specific terms could quantify this trade-off: test whether "EMA AND (daily activities OR functioning)" captures additional relevant papers or merely increases noise. Your own data demonstrate the cost of overly broad terms, 201 irrelevant papers from energy and computer science entered your initial results. The limitations section should acknowledge that excluding acronyms prioritized precision over sensitivity, and that the true scope of EMA literature on daily activities may be larger than your dataset suggests.

Minor concerns

The Venn diagrams visually suggest quantitative proportions but provide no scale or percentages. Either add actual values to each region showing overlap between person, occupation, and environment, or state clearly in the caption that the diagrams are conceptual schematics not drawn to scale.

Summary

Your core finding, that EMA research disproportionately emphasizes personal factors in mental health populations while neglecting occupation and environment, is valuable and timely. The person-environment-occupation lens reveals patterns previous reviews have not examined. Addressing the transparency and data sharing issues above will substantially strengthen the manuscript's impact and ensure it meets publication standards.

For transparency: A large language model was used to improve clarity and concision in this review.**Reviewers' Comments:** Reviewer's Responses to Questions Reviewer's Responses to Questions

**Comments to the Author**

1. Does this manuscript meet PLOS Digital Health’s publication criteria? Is the manuscript technically sound, and do the data support the conclusions? The manuscript must describe methodologically and ethically rigorous research with conclusions that are appropriately drawn based on the data presented.? Is the manuscript technically sound, and do the data support the conclusions? The manuscript must describe methodologically and ethically rigorous research with conclusions that are appropriately drawn based on the data presented.? Is the manuscript technically sound, and do the data support the conclusions? The manuscript must describe methodologically and ethically rigorous research with conclusions that are appropriately drawn based on the data presented.? Is the manuscript technically sound, and do the data support the conclusions? The manuscript must describe methodologically and ethically rigorous research with conclusions that are appropriately drawn based on the data presented.

Reviewer #1: Partly

Reviewer #2: Yes

2. Has the statistical analysis been performed appropriately and rigorously?

Reviewer #1: N/A

Reviewer #2: Yes

3. Have the authors made all data underlying the findings in their manuscript fully available (please refer to the Data Availability Statement at the start of the manuscript PDF file)?

The PLOS Data policy requires authors to make all data underlying the findings described in their manuscript fully available without restriction, with rare exception. The data should be provided as part of the manuscript or its supporting information, or deposited to a public repository. For example, in addition to summary statistics, the data points behind means, medians and variance measures should be available. If there are restrictions on publicly sharing data—e.g. participant privacy or use of data from a third party—those must be specified.requires authors to make all data underlying the findings described in their manuscript fully available without restriction, with rare exception. The data should be provided as part of the manuscript or its supporting information, or deposited to a public repository. For example, in addition to summary statistics, the data points behind means, medians and variance measures should be available. If there are restrictions on publicly sharing data—e.g. participant privacy or use of data from a third party—those must be specified.requires authors to make all data underlying the findings described in their manuscript fully available without restriction, with rare exception. The data should be provided as part of the manuscript or its supporting information, or deposited to a public repository. For example, in addition to summary statistics, the data points behind means, medians and variance measures should be available. If there are restrictions on publicly sharing data—e.g. participant privacy or use of data from a third party—those must be specified.requires authors to make all data underlying the findings described in their manuscript fully available without restriction, with rare exception. The data should be provided as part of the manuscript or its supporting information, or deposited to a public repository. For example, in addition to summary statistics, the data points behind means, medians and variance measures should be available. If there are restrictions on publicly sharing data—e.g. participant privacy or use of data from a third party—those must be specified.

Reviewer #1: Yes

Reviewer #2: Yes

4. Is the manuscript presented in an intelligible fashion and written in standard English?

Reviewer #1: Yes

Reviewer #2: Yes

5. Review Comments to the Author

Reviewer #1: While the topic is relevant and timely, following issues need to be addressed

Comment 1: The stated objectives and methodology are partly aligned, as the objective aims to "systematically map the use of EMA to understand lived health" but the methods only perform keyword-based classification of publications rather than analyzing actual EMA applications.

Comment 2: The methodology is not clearly and completely reported. The objectives, measures and unit of analysis and the analysis details are incompletely reported. Also, Which component of R package was used is not mentioned. In addition, why the database WoS selected needs justification.

Comment 3: No manual screening process was implemented to verify relevance of the 3,692 included publications, and the post-hoc exclusion of 343 papers (9.3% of the dataset) for "energy sustainability" and "computer science" topics confirms that irrelevant content was initially included.

Comment 4: The inclusion of irrelevant publications and subsequent selective exclusion raises serious concerns about the validity of all reported findings, as the classification of health categories and PEO components may be systematically biased by contaminated data.

Comment 5: The analysis consists only of descriptive keyword frequencies and basic percentages without employing any advanced bibliometric methods.

Comment 6: The discussion on practical implications of the findings needs to be added.

Recommendation: REJECT

Reviewer #2: Summary:

In the paper entitled “Tracking daily activities with ecological momentary assessment: a bibliometric an

alysis of current use in health”, the authors examined the field of Ecological Momentary Assessments in a bibliometric analysis. The authors describe the importance of everyday research in the context of the WHO and other important organizations. They then highlight the importance of Ecological Momentary Assessments in this context and conduct a bibliometric analysis for this purpose. The authors describe the analysis and the areas analyzed. The results are presented and discussed, including a discussion of the limitations. The authors come to the following conclusion:

"This bibliometric analysis shows that EMA research focuses mainly on personal factors and hardly considers occupational or environmental contexts. Most studies (75%) deal with mental disorders, reflecting the psychological origins of EMA. Broader application in the fields of neurology, cardiology, and oncology could better capture the dynamics of health and everyday activities in the real world."

First, I would like to highlight the positive aspects of the article:

- The article is very well written.

- The article makes a contribution.

- The article discusses relevant background work.

- The article fits the journal's subject area.

- The article addresses a current topic.

- The title of the article matches the content.

The article is relevant and well written, but there are some aspects that could be improved:

- Which module was used for the calculations in R?

- Was other software considered for the bibliometric analysis?

- The discussion does not sufficiently take into account existing work (including the introduction). Here are some examples:

(1) https://www.frontiersin.org/journals/psychiatry/articles/10.3389/fpsyt.2024.1300739/full

(2) https://www.mdpi.com/1424-8220/24/2/472

- “Typical” bibliometric analyses also include aspects such as countries, organizations, etc. Why were these not taken into account?

- Should patient-reported outcome measures not be included in the search or mentioned as a limitation?

- Did any work have to be excluded from the search?

6. PLOS authors have the option to publish the peer review history of their article (what does this mean?). If published, this will include your full peer review and any attached files.). If published, this will include your full peer review and any attached files.). If published, this will include your full peer review and any attached files.). If published, this will include your full peer review and any attached files.

**Do you want your identity to be public for this peer review?** For information about this choice, including consent withdrawal, please see our Privacy Policy....

Reviewer #1: No

Reviewer #2: **Yes:**Rüdiger PryssRüdiger PryssRüdiger PryssRüdiger Pryss

  **Figure resubmission:** While revising your submission, we strongly recommend that you use PLOS’s NAAS tool (https://ngplosjournals.pagemajik.ai/artanalysis) to test your figure files. NAAS can convert your figure files to the TIFF file type and meet basic requirements (such as print size, resolution), or provide you with a report on issues that do not meet our requirements and that NAAS cannot fix. While revising your submission, we strongly recommend that you use PLOS’s NAAS tool (https://ngplosjournals.pagemajik.ai/artanalysis) to test your figure files. NAAS can convert your figure files to the TIFF file type and meet basic requirements (such as print size, resolution), or provide you with a report on issues that do not meet our requirements and that NAAS cannot fix.

After uploading your figures to PLOS’s NAAS tool - https://ngplosjournals.pagemajik.ai/artanalysis, NAAS will process the files provided and display the results in the "Uploaded Files" section of the page as the processing is complete. If the uploaded figures meet our requirements (or NAAS is able to fix the files to meet our requirements), the figure will be marked as "fixed" above. If NAAS is unable to fix the files, a red "failed" label will appear above. When NAAS has confirmed that the figure files meet our requirements, please download the file via the download option, and include these NAAS processed figure files when submitting your revised manuscript. **Reproducibility:** To enhance the reproducibility of your results, we recommend that authors of applicable studies deposit laboratory protocols in protocols.io, where a protocol can be assigned its own identifier (DOI) such that it can be cited independently in the future. Additionally, PLOS ONE offers an option to publish peer-reviewed clinical study protocols. Read more information on sharing protocols at https://plos.org/protocols?utm_medium=editorial-email&utm_source=authorletters&utm_campaign=protocols To enhance the reproducibility of your results, we recommend that authors of applicable studies deposit laboratory protocols in protocols.io, where a protocol can be assigned its own identifier (DOI) such that it can be cited independently in the future. Additionally, PLOS ONE offers an option to publish peer-reviewed clinical study protocols. Read more information on sharing protocols at https://plos.org/protocols?utm_medium=editorial-email&utm_source=authorletters&utm_campaign=protocols

---

## [Editor Report · Decision Letter 1]

18 Feb 2026

PDIG-D-25-00686R1

Tracking  daily  activities  with  ecological  momentary  assessment:  a  bibliometric  analysis  of  current  use  in  health

PLOS Digital Health

Dear Dr. Delooz,Thank you for submitting your manuscript to PLOS Digital Health. After careful consideration, we feel that it has merit but does not fully meet PLOS Digital Health's publication criteria as it currently stands. Therefore, we invite you to submit a revised version of the manuscript that addresses the points raised during the review process.

* A letter that responds to each point raised by the editor and reviewer(s). You should upload this letter as a separate file labeled 'Response to Reviewers'. This file does not need to include responses to any formatting updates and technical items listed in the 'Journal Requirements' section below.'. This file does not need to include responses to any formatting updates and technical items listed in the 'Journal Requirements' section below.'. This file does not need to include responses to any formatting updates and technical items listed in the 'Journal Requirements' section below.'. This file does not need to include responses to any formatting updates and technical items listed in the 'Journal Requirements' section below.

* A marked-up copy of your manuscript that highlights changes made to the original version. You should upload this as a separate file labeled 'Revised Manuscript with Track Changes'.'.'.'.

* An unmarked version of your revised paper without tracked changes. You should upload this as a separate file labeled 'Manuscript'.'.'.'.

We look forward to receiving your revised manuscript.

Kind regards,

Torleif Markussen Lunde

Guest Editor

PLOS Digital Health

Louise AC Millard, PhD

Section Editor

PLOS Digital Health

Leo Anthony Celi

Editor-in-Chief

PLOS Digital Health

orcid.org/0000-0001-6712-6626

**Journal Requirements:**

**Additional Editor Comments (if provided):**

You have done a great job addressing the previous concerns, particularly regarding data availability and methodological transparency. This significantly strengthens the reproducibility of this work.

However, there is one remaining inconsistency that must be corrected before publication:

In the Response to Reviewers (Comment 12 and 21), you state that 343 publications (9.3%) were identified as irrelevant and excluded during the analytical filtering process. However, Figure 2 (Flowchart), the Results text (Section 3.2), and Figure 3 all indicate that 201 publications were excluded for being outside the core scope of disability and health.

Please verify the correct number

And one minor formatting issue: In the "Results of the PEO Components" section, the text now refers to Figure 4 , but the conceptual Venn diagrams are labeled as Figure 5.

Otherwise; great job!

**Reviewers' Comments:**

 **Figure resubmission:**While revising your submission, we strongly recommend that you use PLOS’s NAAS tool (https://ngplosjournals.pagemajik.ai/artanalysis) to test your figure files. NAAS can convert your figure files to the TIFF file type and meet basic requirements (such as print size, resolution), or provide you with a report on issues that do not meet our requirements and that NAAS cannot fix. While revising your submission, we strongly recommend that you use PLOS’s NAAS tool (https://ngplosjournals.pagemajik.ai/artanalysis) to test your figure files. NAAS can convert your figure files to the TIFF file type and meet basic requirements (such as print size, resolution), or provide you with a report on issues that do not meet our requirements and that NAAS cannot fix. 

After uploading your figures to PLOS’s NAAS tool - https://ngplosjournals.pagemajik.ai/artanalysis, NAAS will process the files provided and display the results in the "Uploaded Files" section of the page as the processing is complete. If the uploaded figures meet our requirements (or NAAS is able to fix the files to meet our requirements), the figure will be marked as "fixed" above. If NAAS is unable to fix the files, a red "failed" label will appear above. When NAAS has confirmed that the figure files meet our requirements, please download the file via the download option, and include these NAAS processed figure files when submitting your revised manuscript.**Reproducibility:**To enhance the reproducibility of your results, we recommend that authors of applicable studies deposit laboratory protocols in protocols.io, where a protocol can be assigned its own identifier (DOI) such that it can be cited independently in the future. Additionally, PLOS ONE offers an option to publish peer-reviewed clinical study protocols. Read more information on sharing protocols at https://plos.org/protocols?utm_medium=editorial-email&utm_source=authorletters&utm_campaign=protocolsTo enhance the reproducibility of your results, we recommend that authors of applicable studies deposit laboratory protocols in protocols.io, where a protocol can be assigned its own identifier (DOI) such that it can be cited independently in the future. Additionally, PLOS ONE offers an option to publish peer-reviewed clinical study protocols. Read more information on sharing protocols at https://plos.org/protocols?utm_medium=editorial-email&utm_source=authorletters&utm_campaign=protocols

---

## [Editor Report · Decision Letter 2]

12 Mar 2026

Tracking  daily  activities  with  ecological  momentary  assessment:  a  bibliometric  analysis  of  current  use  in  health

PDIG-D-25-00686R2

Dear ms. Delooz,

We are pleased to inform you that your manuscript 'Tracking  daily  activities  with  ecological  momentary  assessment:  a  bibliometric  analysis  of  current  use  in  health' has been provisionally accepted for publication in PLOS Digital Health.

Best regards,

Torleif Markussen Lunde

Guest Editor

PLOS Digital Health